# Super-Responders in Moderate–Severe Psoriasis under Guselkumab Treatment: Myths, Realities and Future Perspectives

**DOI:** 10.3390/life12091412

**Published:** 2022-09-10

**Authors:** Ricardo Ruiz-Villaverde, Fiorella Vasquez-Chinchay, Lourdes Rodriguez-Fernandez-Freire, Jose C. Armario-Hita, Amalia Pérez-Gil, Manuel Galán-Gutiérrez

**Affiliations:** 1Dermatology Department, Hospital Universitario San Cecilio, Avda Conocimiento 33, 18016 Granada, Spain; 2Dermatology Department, Hospital Quirón Salud Sagrado Corazón, 41013 Sevilla, Spain; 3Dermatology Department, Hospital Universitario Virgen del Rocio, 41013 Sevilla, Spain; 4Dermatology Department, Hospital Universitario Puerto Real, 11510 Cádiz, Spain; 5Dermatology Department, Hospital Universitario Virgen de Valme, 41014 Sevilla, Spain; 6Dermatology Department, Hospital Universitario Reina Sofía, 14004 Córdoba, Spain

**Keywords:** guselkumab, psoriasis, super-responder

## Abstract

A fast skin clearance is the main goal to achieve in psoriasis treatment. Patients that present a fast and exceptional improvement with treatment are called super-responders (SR). There is no consensus on the definition of SR with respect to psoriasis. Included herein is a retrospective analysis of a multicenter, observational study of real clinical practices including patients with moderate-to-severe plaque PSO undergoing treatment with Guselkumab (GUS). This cross-sectional analysis includes information on patients between February 2019 to February 2022. A SR is a patient that achieved a PASI = 0 at weeks 12 and 24. Analyses have been performed “as observed” using GraphPad Prism version 8.3.0 for Windows (GraphPad Software, San Diego, CA, USA, At baseline, the PASI is significantly correlated with VAS_pruritus, BSA, and DLQI, while DLQI is significantly correlated with VAS_pruritus. Significant correlations increase in number and magnitude over the follow-up time. In relation to the univariate logistic models carried out, only three variables showed a significant association with the super-responder variable: depression, VAS_pruritus, and DLQI.SR patients, who show a faster evolution in PASI and BSA improvement than non-SRs. Based on the results obtained, it would be possible to also include DLQI and VAS_pruritus in the broader concept of the SR.

## 1. Introduction

A fast skin clearance is the main goal to achieve in psoriasis treatment. Patients that present a fast and exceptional improvement with treatment are called super-responders (SR). This concept is not a unique feature of psoriasis; it is used in severe asthma [1], cancer [2], and cardiac resynchronization therapies [3], among others.

The management of different immune-mediated dermatological diseases through biological therapy with a high economic impact has resulted in the identification of patient profiles that respond more quickly and sustainably to treatments. Therefore, the ultimate goal of characterizing SR patients acquires pharmacoeconomic overtones, with the aim of optimizing the treatment of those patients whose characteristics allow for an optimal and sustained response over time.

In patients with chronic spontaneous urticaria (CSU), two monitoring indexes have been used to adjust the degree of response: UAS7 (the Urticaria activity index) and UCT (the Urticaria control test) [4]. The first of them has been more commonly used by dermatologists, where an excellent degree of response for UAS7 = 0. Interestingly, the concept of the SR linked to treatment with omalizumab has not been explicitly investigated. The focus of interest is linked to slow responders, rapid responders, or non-responders [5,6]. Accordingly, clinical biomarkers [7] and analytical biomarkers have been studied that try to classify patients in one or another group. The factors that have been associated with a worse prognosis in terms of CSU activity and its episodes have been studied by Curto et al. [8] in a large series in Spain, reporting results for multiple episodes of CSU (19.2% had more than one lifetime episode of CSU), late-onset (63.6% of patients developed first their onset of CSU after the age of 45 years), concomitant CIndU (Chronic inducible Urticaria) (20.2%), and functional serum autoreactivity. However, those patients with the best responses have not been characterized, so in an interpretive analysis those who do not meet the aforementioned characteristics can be considered good or optimal responders.

In hidradenitis suppurativa, the concept of SR has not been explored either and we have only found an approximation of a conceptualization in the literature. Cao et al. [9] define super-responder patients as those patients who reach 75% improvement in the development of inflammatory nodules (AN) without the appearance of new abscesses or fistulas in relation to baseline. An improvement in the HiSCR activity index is not considered indicative in the definition of SR. Post-treatment reductions of plasma concentrations of the C-X-C motif chemokine ligand (CXCL)9, CXCL8 (interleukin-8), and CCL19 (macrophage inflammatory protein 3β) were greater in adalimumab super-responders than in non-responders (*p* = 0.026, *p* = 0.044 and *p* = 0.026, respectively), opening a new path in its analytical determination to characterize patients with an optimal response [9].

Ruiz-Villaverde et al. [10] have recently reviewed the concept of the SR as a future perspective in patients with moderate–severe atopic dermatitis. The definition of SR with respect to AD was first established by the Dutch registry of AD published by Ariëns et al. [11] (2020). The goal of this study was to evaluate the 52-week effectiveness and safety of dupilumab in a prospective, multicenter cohort of adult patients with treatment-refractory AD. This group established the three key domains that are then systematically repeated to identify a patient as a SR: an improvement in signs (EASI score), symptoms (peak pruritus NRS), and QoL (DLQI score). These criteria have been replicated in all five publications we have reviewed [11,12,13,14,15] with slight differences among them. AD is possibly the only immune-mediated dermatological disease that includes patient-reported outcomes (PROs) within the definition of SR.

There is no consensus on the definition of a SR with respect to psoriasis; however, this term has been used in different studies [16,17,18]. In 2020, Reich and collaborators presented a descriptive poster to the *American College of Dermatology Conference* where they defined a population of SR patients with respect to guselkumab [18]. This sub-analysis of Voyage 1 and 2 defined SRs as patients that achieved a PASI100 response at either week 20 or 28. They presented greater PASI75, PASI90, and PASI100 responses over 16 weeks than the regular population. The baseline demographics indicated that they were lighter (weight ≤ 90 kg [62.4% vs. 51.4%]), less obese (body mass index ≥ 30 kg/m^2^ [35.4% vs. 45.3%]), had a lower baseline disease severity (Investigator’s Global Assessment score of 4 [16.2% vs. 26.5%]), and had less previous use of systemic non-biologic treatments (60.9% vs. 69.0%) [18].

This definition and those of previous findings must be able to be evaluated in real clinical practice since SRs are a population of patients that are not included in clinical trials on a regular basis, as the criteria are more restrictive.

The following objectives have been set in the present study:To study the clinical and demographic differences in super-responders and in those who are not;To analyze the differences in the variables under study (PASI—Psoriasis activity skin index, BSA—Body surface area-, DLQI—Dermatology life quality index, and VAS—Visual analogic scale-pruritus) between both groups of patients (SR/non-SR) throughout the study period (baseline, 12, 24, 36, 52, 76 and 104 weeks);To perform a multivariate logistic analysis to find the baseline demographic and pathological characteristics that may be associated with the response of super-responders.

## 2. Material and Methods

### 2.1. Study Design

This is a retrospective analysis of a multicenter, observational study of real clinical practices including patients with moderate-to-severe plaque PSO undergoing treatment with GUS. This cross-sectional analysis includes information of patients between February 2019 to February 2022. A total of six tertiary hospitals in Andalusia (Spain) participated in this study. This study has been approved by the Ethics Committee of Hospital Universitario San Cecilio (HUSC-DERM-2022_006). Before inclusion into the study, patients gave their informed consent.

### 2.2. Patients

Inclusion criteria were:(1)Adult moderate-to-severe plaque PSO patients;(2)PSO diagnosis since ≥1 year;(3)Treatment with Guselkumab (GUS) > 24 weeks. Patients received GUS following data sheet specifications (100 mg administered subcutaneously at weeks 0 and 4, followed by maintenance every 8 weeks).

Exclusion criteria were:(1)Other types of PSO different than psoriasis vulgaris;(2)Inability to sign the informed consent document.

### 2.3. Definition of Superresponder

SR is a patient that achieves PASI = 0 at weeks 12 and 24.

Non-SR is a patient that does not achieve PASI = 0 at weeks 12 and 24.

The study period includes different temporary endpoints (baseline, 12, 24, 36, 52, 76, and 104 weeks).

### 2.4. Statistical Analysis

Analyses have been performed “as observed” using GraphPad Prism version 8.3.0 for Windows (GraphPad Software, San Diego, CA, USA, www.graphpad.com, accessed on 20 March 2022).

A descriptive analysis has been carried out for both the response variables evaluated throughout the follow-up period and for the clinical and demographic variables. In order to evaluate continuous variables, the mean, standard deviation, and median have been indicated. For categorical variables, absolute and relative frequencies will be presented.

To evaluate the possible differences in the demographic and clinical variables between the two groups of patients (SR versus non-SR), the odds ratio or univariate, as well as their 95% confidence intervals, have been estimated. If the frequency of any cell is less than 1, the Haldane–Ascombe correction will be applied in order to estimate the confidence interval. The association with categorical variables will also be tested using the *X*^2^ test, with the simulated *p*-value referring to those variables that have at least 5% of cells with expected values less than 5.

Regarding continuous variables, the association was evaluated using the t-test or the Mann–Whitney–Wilcoxon test if the variables were not normal. The selected alpha value is 0.05 (*p*-values less than this value are considered significant results). For continuous variables, the association was evaluated using the *t*-test or the Mann–Whitney–Wilcoxon test if the variables were not normal. The selected alpha value is 0.05 (*p*-values less than this value are considered significant results). Finally, the predictive capacity of the clinical and demographic variables collected at baseline on the probability of being a super responder was evaluated using logistic regression models (set at 52 weeks, since from this predictive point the number of missing values prevents the calculations from being carried out reliably).

## 3. Results

Table 1 presents the descriptive analysis of the demographic and clinical variables with respect to whether or not a patient is a super-responder. In our cohort, 28 patients met the SR criteria and 72 were classified as non-SRs.

At baseline, the PASI (Psoriasis activity skin index) is significantly correlated with the VAS_pruritus (Visual analogic scale of pruritus), BSA (Body surface area), and DLQI (Dermatology quality life index), while the DLQI is significantly correlated with the VAS_pruritus (Table 2). Significant correlations increase in number and magnitude over the follow-up time.

A total of 100 patients were included in the analysis: 29% had previously received anti-TNFα (*n* = 29; ADA, ADA-biosimilar, and etanercept), 29% anti-IL17 (*n* = 29; SEC and IXE), and 42% anti-IL12/23 (*n* = 42; UST). A total of nine discontinuations were reported throughout the study: in the anti-TNFα group, three patients discontinued due to a lack of effectivity; the anti-IL17 group presented two secondary failures; and the anti-IL12/23 group presented one primary failure, one secondary failure, and one adverse event.

In relation to the univariate logistic models carried out, only three variables show a significant association with the super-responder variable: depression, VAS_pruritus, and DLQI. We have not found statistical differences when segmenting both groups into age segments and analyzing the impact of age on being a super-responder. Table 3 shows the odds ratio and the 95% confidence interval of the variables with a significant association.

(a)Patients with depression have reduced odds of being a super-responder by a factor of 0.182 (risk reduction of 82%, although the confidence interval is very wide);(b)For each unit increase in the VAS_pruritus, the odds of being a super-responder are reduced by a factor of 0.826 (17.4% reduction);(c)For each unit increase in the DLQI, the odds of being a super-responder are reduced by a factor of 0.909 (9.1% reduction).

Figure 1 shows the results of the longitudinal regression models that evaluated the temporal evolution of the PASI, BSA, VAS-pruritus, and DLQI variables, depending on whether or not the patients are super-responders.

Regarding the four variables, there are significant differences between super-responders and non-super-responders, as well as significant differences in the evolution of the values of said variables. The interaction between time and the super-responder variable is not significant (that is, the pattern of evolution is not significantly different in both groups). As we can see in the evolution graphs, the values of the variables are markedly reduced in the first 4 weeks, but stabilize later, except for the variable VAS-pruritus, where there is an increase in week 12.

## 4. Discussion

The first approximation of the concept of a SR in psoriasis is that established by Talamonti [16], who established that patients with an HLA-C* 06:02 present a better level of therapeutic response to ustekinumab in a study of *n* = 164 patients, where survival was established from a group of patients positive for this biomarker versus a group that was not. In this study, the SR patients did not present a predominance in terms of sex, they were significantly younger, possessed a lower BMI, were younger upon their initial diagnosis but experienced a longer duration of the disease, and had significantly lower PASI scores. Although a proper definition has not been established, the authors confirm, based on previous analyses [19,20], that the retention rate of ustekinumab at 2 years was 27% higher in the super-responder group of patients.

A first approach in daily clinical practice was our first point of research for categorizing SRs. We defined a SR as a patient that achieves a PASI = 0 at weeks 12 and 24 [21]. Of the 87 individuals included in our work, 16% were SRs. We did not find differences in baseline demographics between SRs and non-SRs. However, SRs seemed to be younger, more bio-naïve, presented shorter psoriasis durations, and fewer comorbidities (such as diabetes, arterial hypertension, and dyslipidemia). Our SR patients also achieved greater PASI100, PASI90, and PASI75 responses over 52 weeks. We identified similar results for other parameters such as BSA, Pruritus, and DLQI, and the drug survival rate proved to be better than non-SR, without significancy.

Currently, the role of SRs in the maintenance of a response to guselkumab through the GUIDE study is being studied [22]. This is a phase 3b, randomized, double-blind, and multicenter study comparing the treatment effects of guselkumab in patients with short (≤2 years) or long (>2 years) durations of plaque-type psoriasis, measured from the first appearance of psoriatic plaques. They defined an SR as a patient that achieves a PASI = 0 at weeks 20 and 28 [22]. Significantly more patients with short disease durations were classified as SRs [23], and the use of previous biologics was negatively associated with the probability of achieving an SR response [24]. These two results highlight the importance of an early intervention. In different clinical studies, the direct inhibition of IL-23 has been shown to successfully induce remission in psoriasis with clearance rates that exceed those achieved by other biological and non-biological treatments. Approximately 44% of patients achieve a PASI score = 0 and approximately 70% of patients achieve a PASI score ≤ 1 after 24 weeks of treatment with guselkumab [25,26,27]. Thus, this clinical trial tries to discuss the concept of modifying the disease course towards long-term remission as a potential novel treatment goal for psoriasis. In the next few months, it will be investigated whether an optimized posology of guselkumab every 16 weeks could behave similar to the current maintenance every 8 weeks in SR patients [22]. Mechanistic experiments will be carried out to understand the molecular changes behind the outcomes.

The most extensive study in real clinical practices with the aim of characterizing SR patients and their opposites, the patients most refractory to biological treatments, has been published by Loft et al. [28], who evaluated the degree of response of 3280 patients from the Danish psoriasis registry (DERMBIO). In this case, all the patients in the registry have been considered, regardless of the prescribed treatment, and a new definition of an SR has been adopted. In this study, SRs are patients who have only received a biological treatment for at least a period of 5 consecutive years without their absolute PASI having been greater than three in this period after the initial 6 months and until completing the 5 years. For this reason, only patients receiving treatment with TNF-a blockers and anti-IL/23 could be considered SRs, since patients receiving anti-IL17 or 23 treatments did not meet the time requirement. Once again, a criterion of efficacy and activity is taken into account, a PASI < 3 is employed, PROs are not considered, and the concept of persistence or durability of the drug is introduced.

The Danish registry establishes 6.3% of patients as super-responders (which is equivalent to 6.5% of refractory patients). The main differences between both groups are established with the BMI and the number of comorbidities. SR patients have a lower BMI and less of an inflammatory burden (fewer comorbidities) as has been confirmed in other studies [29]. The main difference from an anthropometric and social point of view possibly lies in the better socioeconomic status of SR patients (64% vs. 38%), in line with what was previously communicated [30].

In our actual cohort (*n* = 100), despite not finding statistically significant differences, SR patients are predominantly male, young, possess a lower BMI, and have fewer total comorbidities. These last two characteristics point to a lower baseline inflammatory burden in SR patients. These data show a complete agreement with what has previously published by Loft [28] in the Danish registry, despite the fact that Guselkumab is not one of the drugs included in the determination of a SR status, as we have explained.

There are no great differences in the time of evolution of the disease or the number of previous treatments. The local regulatory authorities in the hospitals participating in the study require the prescription of Guselkumab at least after systemic treatment and a biosimilar TNF-blocker drug in most cases, which makes it difficult to find differences in the times of the evolution of the disease in both subgroups. For this reason, the data presented may show significant differences with those obtained in the recently started GUIDE study [22].

Finally, and as a significant finding, we would wish to emphasize the correlation between the evolution of PASI and the PROs, i.e., the DLQI and VAS_pruritus. The parallel evolution of these four indexes has already been revealed in our cohort by measuring the effectiveness of Guselkumab in the short [31] and medium terms [32], which was related to the better therapeutic adherence of patients to the drug, reaching survival rates greater than 90% upon independence of the drug of origin after switching due to a lack of efficacy or safety reasons [33]. In other immune-mediated diseases such as atopic dermatitis (AD) that began in the study years with biological treatments for its control, the SR concept includes the evaluation of the DLQI and pruritus for its definition [11,12,13,14,15]. It would be interesting to consider a holistic approach to these four parameters to define the future SRs of any biological drug in the treatment of moderate–severe psoriasis.

Some limitations of this study are the fact that is a retrospective study, the presence of unbalanced groups, and, subsequently, the absence of adjustment in clinical and demographic basal characteristics between the groups.

In conclusion, SR patients show a faster and better evolution in PASI and BSA improvement than non-SRs. Based on the results obtained, it would be possible to also include the DLQI and VAS_pruritus in a broader and initial concept of SRs.

## Figures and Tables

**Figure 1 life-12-01412-f001:**
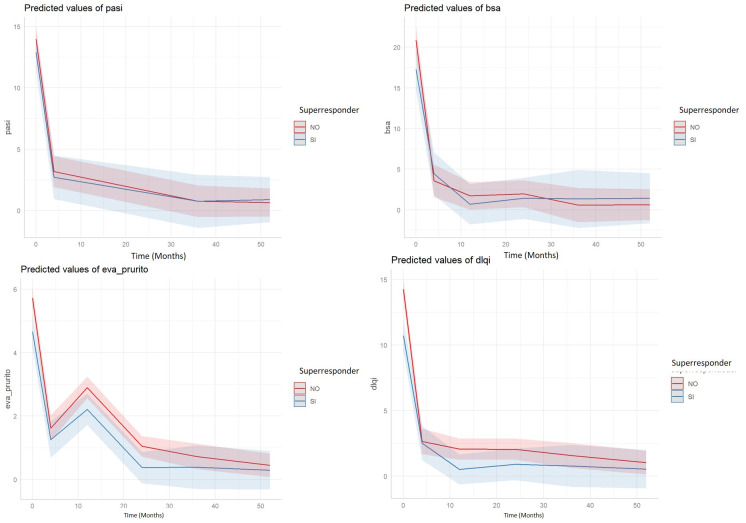
Model-predicted PASI, BSA, VAS pruritus, and DLQI values over 52 weeks of follow-up for super-responders (blue) and non-super-responders (red) (shaded bands represent the 95% prediction interval).

**Table 1 life-12-01412-t001:** Descriptive table of the clinical and demographic variables according to the patient’s responses (*p*.overall: *p*-value of the corresponding association test).

.	*Non-SR*	*SR*	*p* *-* *Value*
	*N = 72*	*N = 28*	
**Sex**			0.692
**Male**	44 (61.1%)	19 (67.9%)	
**Female**	28 (38.9%)	9 (32.1%)	
**Age (Years)**	51.6 (14.8)	47.1 (13.4)	0.153
**Family History of PSO**	12 (23.1%)	7 (31.8%)	0.620
**Time of evolution (Years)**	19.0 [11.0;29.0]	22.5 [11.0;26.8]	0.797
**Weight (kg)**	85.0 [75.0;97.2]	83.5 [69.5;90.2]	0.449
**Height (cm)**	170 [162;178]	175 [169;178]	0.156
**BMI**	29.8 [26.3;32.0]	26.9 [23.5;30.7]	0.062
**Obesity**	35 (48.6%)	8 (28.6%)	0.111
**PsA**	14 (19.7%)	5 (17.9%)	1.000
**Diabetes**	15 (21.1%)	4 (14.3%)	0.621
**Hypertension**	19 (26.8%)	6 (21.4%)	0.769
**Dyslipidemia**	28 (39.4%)	10 (35.7%)	0.910
**Depression:**	12 (16.9%)	1 (3.57%)	0.102
**Nonalcoholic Fatty Liver**	9 (12.7%)	4 (14.3%)	1.000
** *n* ** **_previous systemic treatments:**			0.494
**0**	3 (4.17%)	0 (0.00%)	
**1**	10 (13.9%)	6 (21.4%)	
**2**	43 (59.7%)	18 (64.3%)	
**3**	16 (22.2%)	4 (14.3%)	
**n_previous biological treatments**	2.00 [1.00;3.00]	2.00 [1.00;3.00]	0.664

**Table 2 life-12-01412-t002:** Descriptive table of the PASI, BSA, VAS_pruritus, and DLQI variables evaluated during follow-up, depending on the patient’s response.

.	PASI	*p*-Value	BSA	*p*-Value	VAS Pruritus	*p*-Value	DLQI	*p*-Value
**Basal**	13.0 [8.50;16.0]	0.314	17.0 [8.00;24.0]	0.223	5.00 [3.00;6.00]	0.049	11.0 [5.00;15.0]	0.013
**sem4**	0.60 [0.00;2.75]	0.005	1.00 [0.00;2.75]	0.009	0.00 [0.00;1.00]	0.034	0.00 [0.00;2.00]	0.057
**sem12**	0.00 [0.00;0.00]	<0.001	0.00 [0.00;0.00]	<0.001	2.00 [2.00;2.00]	<0.001	0.00 [0.00;0.00]	<0.001
**sem24**	0.00 [0.00;0.00]	<0.001	0.00 [0.00;0.00]	<0.001	0.00 [0.00;0.00]	<0.001	0.00 [0.00;0.00]	<0.001
**sem36**	0.00 [0.00;0.00]	<0.001	0.00 [0.00;0.00]	<0.001	0.00 [0.00;0.00]	0.044	0.00 [0.00;1.00]	0.043
**sem52**	0.00 [0.00;0.00]	0.001	0.00 [0.00;0.00]	0.001	0.00 [0.00;0.00]	0.044	0.00 [0.00;0.75]	0.044

**Table 3 life-12-01412-t003:** Results of the univariate logistic models (Model.L.R.: likelihood ratio test statistic, df: degrees of freedom, *p*: *p*-value, C: concordance index, and Dxy: Sommer index).

Terms	Model.L.R.	d.f.	*p* Value	C	Dxy	Signif
**Sex**	0.399	1	0.528	0.534	0.067	ns
**Age**	1.958	1	0.162	0.564	0.128	ns
**Familiar History PsO**	0.603	1	0.437	0.544	0.087	ns
**Time of evolution**	0.097	1	0.756	0.517	0.033	ns
**Weight_kg**	0.174	1	0.677	0.549	0.098	ns
**Height_cm**	2.811	1	0.094	0.592	0.183	ns
**BMI**	1.993	1	0.158	0.621	0.242	ns
**Obesity**	3.402	1	0.065	0.600	0.200	ns
**PsA**	0.045	1	0.831	0.509	0.019	ns
**Diabetes**	0.635	1	0.426	0.534	0.068	ns
**Hypertension**	0.309	1	0.578	0.527	0.053	ns
**Dyslipidemia**	0.118	1	0.731	0.519	0.037	ns
** Depression **	3.856	1	0.050	0.567	0.133	
**Nonalcoholic fatty liver**	0.045	1	0.832	0.508	0.016	ns
** *n* ** **_previous biological**	0.592	1	0.441	0.527	0.055	ns
**PASI**	1.553	1	0.213	0.566	0.132	ns
**BSA**	2.132	1	0.144	0.580	0.159	ns
** VAS pruritus **	4.698	1	0.030	0.630	0.260	
** DLQI **	6.891	1	0.009	0.668	0.336	**

** *p* < 0.05.

## Data Availability

Data supporting reported results may be provided upon reasonable request to authors.

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
