# Peer review of "Super-Responders in Moderate–Severe Psoriasis under Guselkumab Treatment: Myths, Realities and Future Perspectives"

_life, 2022, doi:10.3390/life12091412_

Round 1
Reviewer 1 Report
The manuscript entitled “Superresponders in moderate-severe psoriasis under Guselku- 1 mab treatment: Myths, realities and future perspectives” by Ricardo Ruiz-Villaverde et al focuses on the retrospective analysis of a multicentre, observational, study of real clinical practice including patients with moderate-to-severe plaque PSO in treatment with Guselkumab. The This cross-sectional analysis includes information of patients between February 2019 to February 2022.
The method used is adequately described and the results obtained are presented well enough. Besides, the conclusions of the authors are supported by the clear graphs.
The results obtained are significant for a better understanding of effective of treatment the pathology in question and can be used for find of therapy succussed.
The following critical remarks can be made.
1. It is not clear what the authors mean by the VAS, BSA and DLQI. It is necessary to give this information.
2. The authors of the article have not explained the correlation between the SR of patients and their ages. Additional studies involving older patients are necessary.
Author Response
Reviewer 2
Congratulations to the authors for the interesting research, which presents a very important problem in psoriatic patients. The results give an important guidance to clinicians.
Im so glad that you took into account the previous biologic treatment. Would you like to describe this point more. Was the biologic withdrawaled because of lack of efficacy or intolerance? Were the patient treated with other iIL23 before?
The structure of the manuscript is correct, the content is understandable to the reader. Properly conducted examination, well-defined inclusion / exclusion criteria, sufficiently large sample group, described limitations. The authors draw the right conclusions from the work on research. I read the manuscript with great pleasure.
Author reply: Thank you very much for your kind words. It is a manuscript worked with much love and dedication. We have completed the information requested by the editor. None of the patients had previously received another anti-IL23 drug since Guselkumab was the first drug of its kind marketed in our country.
Reviewer 2 Report
Psoriasis is a relatively difficult disease to treat. It is an important means to study the pathogenesis of psoriasis and find an effective treatment to solve the suffering of psoriasis patient. Through a retrospective analysis of multicenter and observational studies in real clinical practice (including treatment of GUS in patients with moderate to severe plaque PSO), the authors concluded that they did not find baseline demographic differences between SR and non-SR, SR appears to be younger, more biologically naive, with shorter psoriasis duration and fewer comorbidities (such as diabetes, arterial hypertension, and dyslipidemia), inhibition of IL-23 can effectively relieve the symptoms of psoriasis, and its efficacy is superior to other biological and non- biological treatments, SRs were closely related to VAS_pruritus and DLQI of psoriasis patients, and there were statistically significant differences. In general, this is an interesting work, but the results are not significantly innovative. In particular, the display of the original data, the scientific evaluation of the model used, and the clinical guidance for obtaining the results are not enough. The following are some minor comments.
(1) In abstract, the authors showed that in relation to the univariate logistic models carried out, only three variables show a significant association with the super-responder variable: depression, VAS_pruritus and DLQI. However, in result of table 3, the P value of “Depression” is 0.050.
(2) In table 3, the significance of “VAS pruritus” is not marked.
(3) On lines 18, 32, 27, there is no space after the end of a complete sentence, and there is an extra quotation mark on line 23.
(4) The position of "predicted values of plqi" in Figure 1 should be aligned with the above “Predicted values of bsa”.
Author Response
Psoriasis is a relatively difficult disease to treat. It is an important means to study the pathogenesis of psoriasis and find an effective treatment to solve the suffering of psoriasis patient. Through a retrospective analysis of multicenter and observational studies in real clinical practice (including treatment of GUS in patients with moderate to severe plaque PSO), the authors concluded that they did not find baseline demographic differences between SR and non-SR, SR appears to be younger, more biologically naive, with shorter psoriasis duration and fewer comorbidities (such as diabetes, arterial hypertension, and dyslipidemia), inhibition of IL-23 can effectively relieve the symptoms of psoriasis, and its efficacy is superior to other biological and non- biological treatments, SRs were closely related to VAS_pruritus and DLQI of psoriasis patients, and there were statistically significant differences. In general, this is an interesting work, but the results are not significantly innovative. In particular, the display of the original data, the scientific evaluation of the model used, and the clinical guidance for obtaining the results are not enough. The following are some minor comments.
Author reply: Thank you very much for your kind words. We have performed this manuscript with much love and dedication. We understand that the methodology may not always be completely convincing, but we do believe that it provides evidence to continue working in this line.
- In abstract, the authors showed that in relation to the univariate logistic models carried out, only three variables show a significant association with the super-responder variable: depression, VAS_pruritus and DLQI. However, in result of table 3, the P value of “Depression” is 0.050.
Author reply: On methods we have stated: p-values equal to or less than this value will be considered significant .
- In table 3, the significance of “VAS pruritus” is not marked.
Author reply: We are sorry that we do not agree with the reviewer since the p value of VAS pruritus, depression and DLQI are marked in red
- On lines 18, 32, 27, there is no space after the end of a complete sentence, and there is an extra quotation mark on line 23.
Author reply:All grammar mistakes have been reviewed.
- The position of "predicted values of plqi" in Figure 1 should be aligned with the above “Predicted values of bsa”.
Author reply: The figure 1 have been replaced and de predicted values of DLQI and BSA aligned.
Reviewer 3 Report
Congratulations to the authors for the interesting research, which presents a very important problem in psoriatic patients. The results give an important guidance to clinicians.
Im so glad that you took into account the previous biologic treatment. Would you like to describe this point more. Was the biologic withdrawaled because of lack of efficacy or intolerance? Were the patient treated with other iIL23 before?
The structure of the manuscript is correct, the content is understandable to the reader. Properly conducted examination, well-defined inclusion / exclusion criteria, sufficiently large sample group, described limitations. The authors draw the right conclusions from the work on research. I read the manuscript with great pleasure.
Author Response
Granada 25th August 2022
Comment to reviewers
Reviewer 1
The manuscript entitled “Superresponders in moderate-severe psoriasis under Guselku- 1 mab treatment: Myths, realities and future perspectives” by Ricardo Ruiz-Villaverde et al focuses on the retrospective analysis of a multicentre, observational, study of real clinical practice including patients with moderate-to-severe plaque PSO in treatment with Guselkumab. The This cross-sectional analysis includes information of patients between February 2019 to February 2022.
The method used is adequately described and the results obtained are presented well enough. Besides, the conclusions of the authors are supported by the clear graphs.
The results obtained are significant for a better understanding of effective of treatment the pathology in question and can be used for find of therapy succussed.
The following critical remarks can be made.
1.It is not clear what the authors mean by the VAS, BSA and DLQI. It is necessary to give this information.
Author reply: Thank you very much for your kind words. We have made this manuscript with much love and dedication.
In relation to the first point, we have described the abbreviations of the acronyms in the material and methods section as well as in the results section. This is basic terminology of measures of efficacy (PASI and BSA) and PROs (patient reported results) that are measured in all randomized clinical trials and real clinical practice series.
- The authors of the article have not explained the correlation between the SR of patients and their ages. Additional studies involving older patients are necessary.
Author reply. The issue pointed out by the reviewer is important, but the sample size of the population of patients older than 65 years has not allowed us to obtain statistically significant conclusions or with signs of significance, but we agree that it is an interesting field to explore.
Reviewer 2
Congratulations to the authors for the interesting research, which presents a very important problem in psoriatic patients. The results give an important guidance to clinicians.
Im so glad that you took into account the previous biologic treatment. Would you like to describe this point more. Was the biologic withdrawaled because of lack of efficacy or intolerance? Were the patient treated with other iIL23 before?
The structure of the manuscript is correct, the content is understandable to the reader. Properly conducted examination, well-defined inclusion / exclusion criteria, sufficiently large sample group, described limitations. The authors draw the right conclusions from the work on research. I read the manuscript with great pleasure.
Author reply: Thank you very much for your kind words. It is a manuscript worked with much love and dedication. We have completed the information requested by the editor. None of the patients had previously received another anti-IL23 drug since Guselkumab was the first drug of its kind marketed in our country.
The new version of the manuscript is implemented with track changes so that the revisions can be observed by the editorial team and reviewers.
Regards.
Round 2
Reviewer 1 Report
In author's response mixed up the reports of the reviewers. This is probably an editor error.
I wrote: "The method used is adequately described and the results obtained are presented well enough. Besides, the conclusions of the authors are supported by the clear graphs.
The results obtained are significant for a better understanding of effective of treatment the pathology in question and can be used for find of therapy succussed.
The following critical remarks can be made.
1. It is not clear what the authors mean by the VAS, BSA and DLQI. It is necessary to give this information.
2. The authors of the article have not explained the correlation between the SR of patients and their ages. Additional studies involving older patients are necessary.
The author replied to: "Congratulations to the authors for the interesting research, which presents a very important problem in psoriatic patients. The results give an important guidance to clinicians.
Im so glad that you took into account the previous biologic treatment. Would you like to describe this point more. Was the biologic withdrawaled because of lack of efficacy or intolerance? Were the patient treated with other iIL23 before?
The structure of the manuscript is correct, the content is understandable to the reader. Properly conducted examination, well-defined inclusion / exclusion criteria, sufficiently large sample group, described limitations. The authors draw the right conclusions from the work on research. I read the manuscript with great pleasure."
Author reply: Thank you very much for your kind words. It is a manuscript worked with much love and dedication. We have completed the information requested by the editor. None of the patients had previously received another anti-IL23 drug since Guselkumab was the first drug of its kind marketed in our country.
Despite the lack of a response to my comments, the necessary changes have been made in the article.
Author Response
The following critical remarks can be made.
- It is not clear what the authors mean by the VAS, BSA and DLQI. It is necessary to give this information.
Author reply: In relation to the first point, we have described the abbreviations of the acronyms in the material and methods section as well as in the results section. This is basic terminology of measures of efficacy (PASI and BSA) and PROs (patient reported results) that are measured in all randomized clinical trials and real clinical practice series.
- The authors of the article have not explained the correlation between the SR of patients and their ages. Additional studies involving older patients are necessary.
Author reply. The issue pointed out by the reviewer is important, but the sample size of the population of patients older than 65 years has not allowed us to obtain statistically significant conclusions or with signs of significance, but we agree that it is an interesting field to explore
Despite the lack of a response to my comments, the necessary changes have been made in the article.
We apologize for the mistake in the confusion we have had in replying to both reviewers.
Regardds